# Compressive Strength Characteristics of Long Tubular Bones after Hyperthermal Ablation

**Denis Pakhmurin** [1,2,3,*], **Viktoriya Pakhmurina** [1], **Alexander Kashin** [4], **Alexey Kulkov** [4], **Igor Khlusov** [2,3], **Evgeny Kostyuchenko** [1,*], **Ivan Sidorov** [5] and **Ilya Anisenya** [6]

1 Laboratory of Acquisition, Analysis and Control of Biological Signals, Tomsk State University of Control Systems and Radioelectronics, 40 Lenina Str., 634050 Tomsk, Russia; pvv@ie.tusur.ru
2 Department of Morphology and General Pathology, Siberian State Medical University, 2 Moskovsky Trakt, 634050 Tomsk, Russia; khlusov63@mail.ru
3 Department of Medical Cybernetics, Siberian State Medical University, 2 Moskovsky Trakt, 634050 Tomsk, Russia
4 Institute of Strength Physics and Materials Science, Siberian Branch of Russian Academy of Sciences (ISPMS SB RAS), 2/4 Akademicheskiy Str., 634055 Tomsk, Russia; kash@ispms.ru (A.K.); 727@ispms.ru (A.K.)
5 Irkutsk Supercomputer Center of SB RAS, 134 Lermontova, 664033 Irkutsk, Russia; ivan.sidorov@icc.ru
6 Department of General Oncology, Cancer Research Institute, Tomsk National Research Medical Center, Russian Academy of Sciences, 634009 Tomsk, Russia; aii@mail.tsu.ru
* Correspondence: pdo@ie.tusur.ru (D.P.); key@fb.tusur.ru (E.K.); Tel.: +7-3822-70-15-29 (E.K.)

**Abstract:** Thermoablation is used in the treatment of tumorous bones. However, little is known about the influence such thermal treatment has on the mechanical properties of bone tissue. The purpose of this work was to study the influence of thermal treatment in a range of 60–100 °C (in increments of 10 °C) on the structural properties of pig femurs using an original approach that involved a periosteal arrangement of heating elements providing gradual dry heating of the bone from its periphery to its center. Heating of freshly extracted bone tissue segments was performed ex vivo using surface heaters of a Phoenix-2 local hyperthermia hardware system. Mechanical testing followed the single-axis compression scheme (traverse velocity of 1 mm/min). In the 60–90 °C range of heating, no attributes of reduced structural characteristics were found in the samples subjected to thermoablation in comparison to the control samples taken from symmetric areas of adjacent cylinders of healthy bones and not subjected to heat treatment. The following statistically significant changes were found as a result of compressing the samples to 100 °C after the heat treatment: reduced modulus of elasticity and increased elastic strain (strain-to-failure), mainly due to increases in plastic deformation. This finding may serve as evidence of a critical ex vivo change in the biomechanical behavior of bone tissues at such temperatures. Thus, ex vivo treatment of bone tissue in the thermal range of 60–90 °C may be used in studies of thermoablation efficiency against tumor involvement of bones.

**Keywords:** bone deformation; tissue heating; tumor treatment; hyperthermia; tubular bones; thermoablation

## 1. Introduction

Hyperthermia has been used in the practical treatment of malignant bone tumors since the middle of the 20th century. A meta-analysis using the term "hyperthermia and bone" in the PubMed database of the National Institute of Health (USA) [1] showed that starting from the 1980s, the number of publications on this topic has been gradually increasing from single articles to 40–60 papers per year by 2021. In total, 1056 publication references were found, the earliest being published in 1949. The influence of thermal treatment on the mechanical properties of bone tissue is still understudied. Only two articles published after 1993 on topics close to that of our research were discovered in the PubMed database using the search term "bone biomechanics and hyperthermia".

There are various methods of treating oncological diseases of bone tissues, such as hyperthermia, and radio-frequency, thermal, and microwave ablation [2–4]. However, direct thermal heating does not lead to a decrease in the frequency of relapses [2]. In addition, there is the problem of ensuring the required ideal distribution of nanoparticles in the biomaterial when heating is provided [4]. The possibilities of radiation therapy are limited by the radioresistance of osteosarcoma [4] and by its destructive effect on the bone marrow [4].

Despite the differences in effectiveness of the above treatment methods and the presence of additional side effects, the problem of the effect of treatment on bone strength is common to all the above methods.

In recent decades, a radical approach adopted for the treatment of bone tumors involves not only directly resecting a tumor-affected bone segment but also its subsequent reconstruction after the treatment with the aim of restoring and maintaining the bone's functions [5,6].

Reconstruction involves bone implants placed in the defect area. A wide range of materials is used in implant manufacture (metals, polymer composites, ceramics, and coral) [7–9]; however, bone tissue remains preferable as it provides a number of advantages: The use of autogenic bone prevents immunologic havoc. In addition, the bone is biologically active as it is capable of initiating regenerative reactions. At the same time, in a situation of tumor involvement of bones, the required volume of bone tissue often exceeds the volume of material that may be obtained from the patient. Bone tumors may be extensive, thus preventing transplantation of autologous bone.

Replacement of impacted tissues involves an autotransplant after thermal treatment [10,11], beginning with the removal of an impacted area of bone tissue followed by thermal sterilization of the area, then its repositioning in the same anatomic location. A disadvantage of this method is a reduction in the strength of treated bones and insufficient (or lacking) osseous fusion between the ex vivo treated fragment and its host bone.

An original technique was proposed by the Tomsk Cancer Research Institute [12], making provisions for intraoperative thermoablation of large bone fragments from tumor-involved tubular bones without their surgical removal. The bone is initially separated from surrounding tissues and is enclosed with special flexible heaters that provide external thermal insulation. Heat is distributed from the periphery to the center, thus allowing heating of the bone throughout its cross-section. The tumor-involved bone keeps its anatomic integrity when this technique is used. Such a technique may be implemented using a Phoenix-2 local hyperthermy equipment package, which has been developed by Tomsk State University of Control Systems and Radioelectronics [13–16]. The Phoenix-2 local hyperthermia device is unique for both local hyperthermia and thermal ablation. It can be used for heating of soft tissues and bones. Phoenix-2 provides heating only by thermal waves (dry warmth). It does not apply any other kind of energy (current, microwaves, etc.) to the patient tissues. A direct current enters the heater then returns to the Phoenix-2. The heater becomes warmer until its temperature reaches the desired level. This level may be from 45 to 100 °C. The heaters may resemble metal needles or rubber plates. In these experiments, we used the latter type of heaters. They were 15 cm long and 5 cm wide and we could wrap experimental cylinders around them. The instrument is capable of maintaining a wide range of temperatures, thus allowing for various modes of thermal treatment. At the same time, in order to determine an optimal thermal mode for intraoperative thermoablation, it is necessary to study regularities in the change of healthy long bone tissues depending on the mode of hyperthermal treatment.

Studies in deformation behavior of surgically removed bone fragments following their hyperthermal treatment by pasteurizing (60 °C), boiling (100 °C), and autoclaving (120 °C) showed a reduction in the bone tissues' mechanical strength when the treatment temperature was above 60 °C [17,18]. In [19], the researchers studied the mechanical properties of bones while taking the time factor into account. A segment of rabbit shin bone was re-implanted to its initial anatomic location following hyperthermal treatment

(autoclaving or pasteurizing). Obtained data showed that with time, thermally treated transplants demonstrated a reduction in mechanical strength. It was also noted that the autoclaving group of samples demonstrated a higher mechanical strength, which was still 10–16% lower than the control. Pasteurized autologous bone transplants showed the lowest mechanical strength in compression tests (40–67% lower than the control group). Thus, different modes of thermal load (e.g., dry or wet heating) seem to exert different influences on the mechanical properties of bone tissue [20].

The goal of this research was to study the influence of dry thermal treatment on the mechanical strength of pig femurs while taking an original approach where heating elements are placed periosteally to provide gradual dry heating of the bone from its periphery to its center.

## 2. Research Methods

The research was conducted on pig femurs, with test samples being cut from the diaphyseal parts of the bones. The high-temperature treatment employed five thermal levels: 60, 70, 80, 90, and 100 °C. All treatments were 1 h long. The heating was conducted using the surface heaters of a Phoenix-2 local hyperthermy equipment complex (PromEl, Tomsk, Russia); the temperature of the heaters was stabilized at a preselected level. Temperature control involved PT-100 thermal sensors placed on the bone surface (under the heater).

Each thermal mode was studied on three bones. From the diaphyseal part of each bone, two adjacent cylinders were cut with a height of 1 cm and diameter of 3–4 cm. One of the cylinders then underwent the high-temperature treatment while another stayed intact (Figures 1 and 2). Samples in the form of 0.8 × 0.5 × 0.5 cm rectangular blocks were cut from symmetrical areas of the bones then divided into two groups: 1, samples cut from the high-temperature treated bone, 9 samples (3 from each bone); 2, control group, 9 samples (3 from each bone). There were separate control groups for each heating temperature (cut from the same bones as the experimental fragments). Control measurements were conducted at a room temperature of 18–22 °C. Cutting was performed with a water-cooled micro-mill.

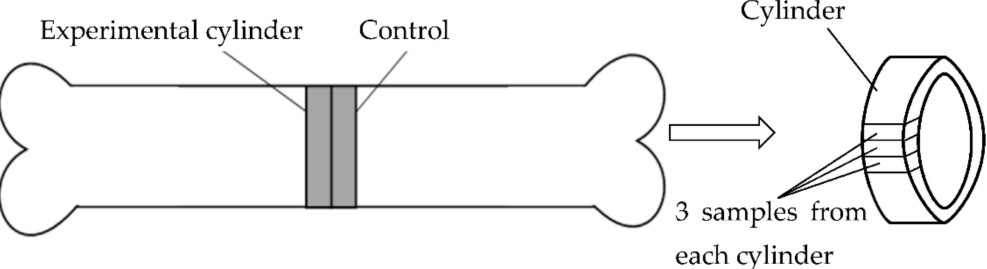

**Figure 1.** Forming cylinders and samples for study.

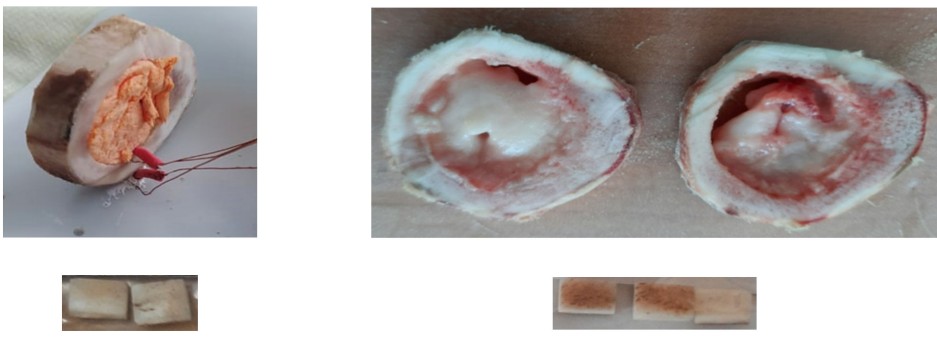

**Figure 2.** Studied cylinders and samples.

Measurement of the strength characteristics of the bone samples in conditions of single-axis compression after thermal ablation was performed using Instron 1185 testing

equipment (ITW Inc., Glenview, IL, USA) [21–23]. The traverse movement speed amounted to 1 mm/min. The samples were placed in such a way that compression was performed along the long axis of the bone, along the osteon direction. The testing stopped at the initial moment of sample destruction. Stress–strain curves were plotted and subsequently used to find the modulus of elasticity. The obtained data underwent statistical treatment using the Statistica 13.3 software package for Windows (TIBCO Software Inc., Palo Alto, CA, USA).

Each temperature level was represented with two independent series (control and hyperthermic treatment), with 9 samples each. As the obtained data demonstrated a large dispersion and took into account variation in the properties of strength and elasticity among various zones of bone cross-sections [24,25], the comparison between experimental and control samples was made not only for the group as a whole but additionally for each individual bone. The normalcy of property distribution was assessed using the Shapiro–Wilk test ($n < 50$) [26]. A significance value of $p < 0.05$ means that the distribution differs from a normal one. When describing the normal distribution of a quantitative property, the mean value and standard deviation are used. The median (Me) and quartile range (Q1–Q3) are used for indicators not conforming to a normal distribution. Assessment of the normalcy of distribution is shown in Table 1.

**Table 1.** Significance values as per the Shapiro–Wilk test.

| Group | Stress-to-Failure | | Strain-to-Failure | | Modulus of Elasticity | | Elastic Deformation | | Plastic Deformation | |
|---|---|---|---|---|---|---|---|---|---|---|
| | Control | Heating | Control | Heating | Control | Heating | Control | Heating | Control | Control |
| 100 °C | 0.46699 | 0.30977 | **0.03275** | 0.05194 | 0.81387 | 0.75861 | 0.48657 | 0.73026 | 0.10561 | 0.06735 |
| 90 °C | 0.66120 | 0.58837 | 0.19796 | **0.00402** | 0.86435 | 0.99101 | 0.19836 | **0.03139** | **0.00422** | **0.00486** |
| 80 °C | 0.23460 | 0.97968 | 0.66224 | 0.14339 | 0.40616 | 0.79511 | 0.05885 | 0.31993 | 0.07168 | 0.28658 |
| 70 °C | 0.17820 | 0.08900 | **0.00861** | **0.04942** | **0.02703** | 0.65011 | 0.17845 | 0.34244 | **0.00040** | **0.00000** |
| 60 °C | 0.67545 | 0.06407 | **0.03088** | **0.01368** | 0.94430 | 0.26179 | 0.31630 | 0.14852 | **0.00028** | **0.00005** |

Note: The values of statistical significance in testing the normal distribution of samples. When $p < 0.05$ (in bold), the null hypothesis of a normal distribution is rejected.

As evident from the table, the Gaussian distribution of the attribute was observed only for the stress-to-failure indicator. Thus, only for this indicator, its mean value was used as an adequate measure of the central tendency in the studied data sample. For the other indicators (strain-to-failure, Young's modulus of elasticity, elastic deformation, and plastic deformation) that showed deviations from the normal distribution, the median value was used to characterize their central trends, as this parameter is less prone to overswings (too large or too small values) that are atypical for the sample as a whole.

In order to assess the significance of the obtained differences between groups, a non-parametric Mann–Whitney test for independent samples was used (U-test) [26].

## 3. Results and Discussion

Before further discussion, it is necessary to show how the different parameters that we used in the study were calculated. The moment of sample destruction corresponded to the maximum withstand force or maximum load. Stress was calculated according to Formula (1):

$$\sigma = \frac{F}{S} \tag{1}$$

where $\sigma$ is stress, MPa; $F$ is stress, N; and $S$ is sample square, mm$^2$.

The calculation of the strain was carried out according to Formula (2):

$$\varepsilon_z = \frac{\Delta h}{h} \tag{2}$$

where $\varepsilon_z$ is strain, dimensionless; $\Delta h$ is the change in sample height, mm; and $h$ is the initial sample height, mm.

In this case, each value of the load $\sigma$ was unambiguously associated with the value of strain $\varepsilon$, which made it possible to graph the stress/strain ($\sigma/\varepsilon$) dependence (see the example in Figure 3). Afterward, the characteristic of the curve was estimated and a straight line was constructed as an approximation of the first steeply increasing section, which has a linear character (Straight-line-1 in Figure 3). Furthermore, there was a point on the graph corresponding to the deviation in the graph from Straight-line-1 by 0.2% (point $\sigma_{0.2}$ in Figure 3). Through this point, a second straight line was drawn parallel to Straight-line-1 (Straight-line-2 in Figure 3). The point corresponding to the maximum stress that the specimen can withstand before fracture is designated in Figure 3 as $\sigma_{max}$ (stress-to-failure). The ordinate of this point corresponds to the stress-to-failure, and the abscissa corresponds to the strain-to-failure.

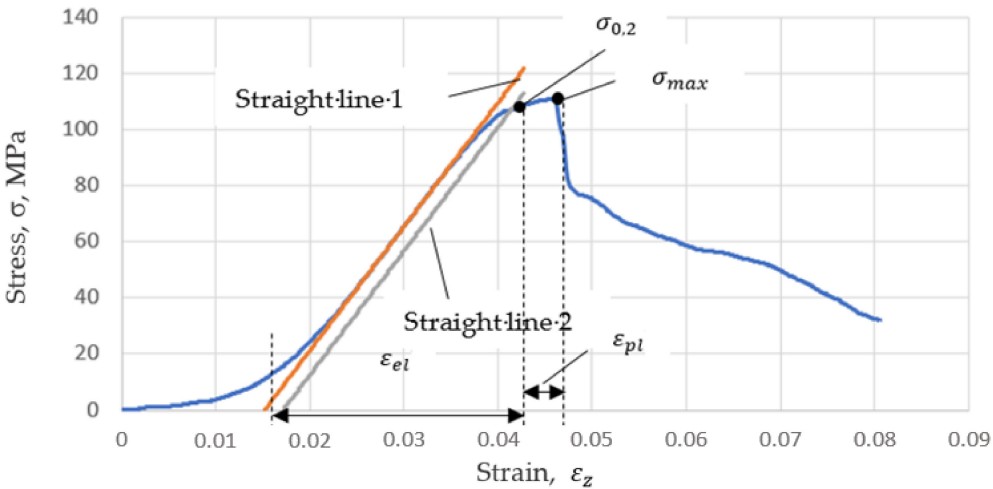

**Figure 3.** An example of the stress/strain graph.

Another parameter that was calculated on the basis of the obtained data was the modulus of elasticity, which was calculated as the tangent of the slope of Straight-line-1 or, taking into account that the straight line can be specified by a formula of the form $y = kx + b$, where the slope angle corresponds to the value of the coefficient $k$.

The difference between the abscissas of the point $\sigma_{0.2}$ and the point of intersection of Straight-line-1 with the abscissa axis corresponds to elastic deformation $\varepsilon_{el}$, and the difference between the abscissas of the points $\sigma_{max}$ and $\sigma_{0.2}$ corresponds to plastic deformation $\varepsilon_{pl}$.

First, deformation curves of the samples heated to 60–100 °C were compared with the control curves (Figures 4–8). Each diagram shows deformation curves for an experimental group and a corresponding control group, as well as trend lines. Experimental curves were approximated with a sixth-degree polynomial for all the hyperthermy ranges.

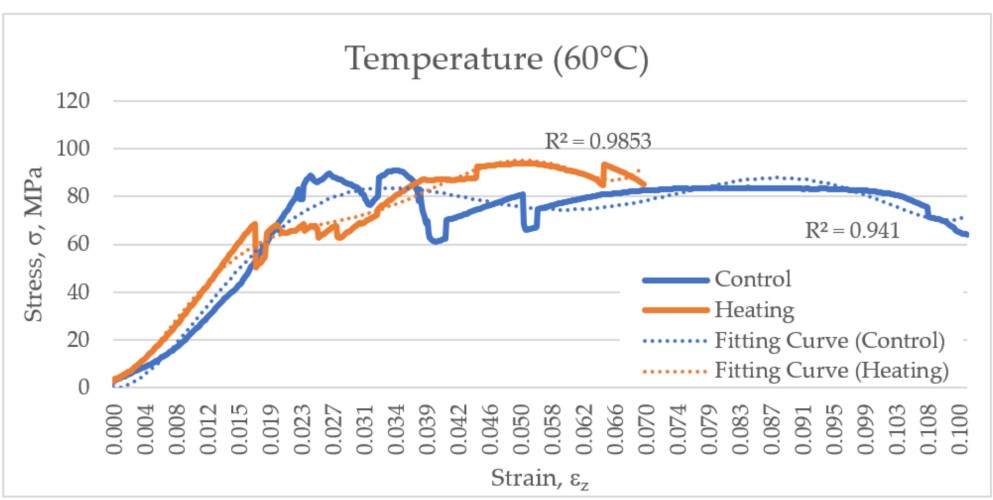

**Figure 4.** Bone samples' deformation curves following their heating to 60 °C.

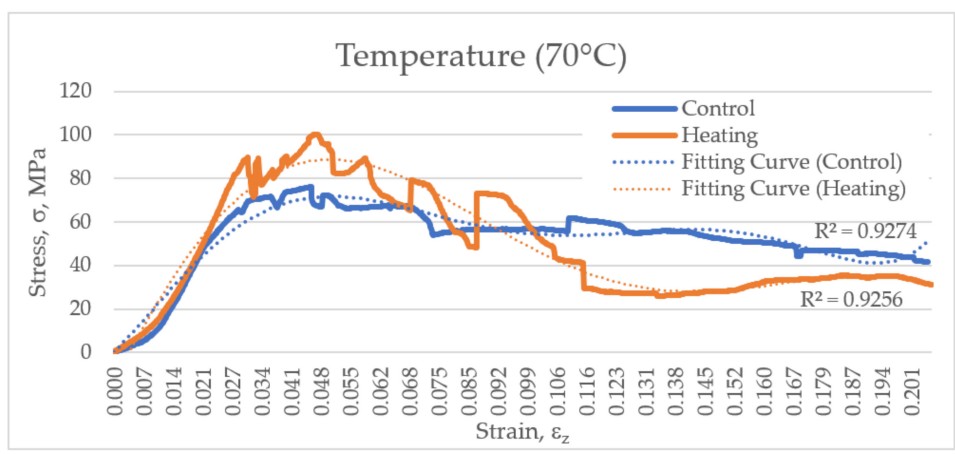

**Figure 5.** Bone samples' deformation curves following their heating to 70 °C.

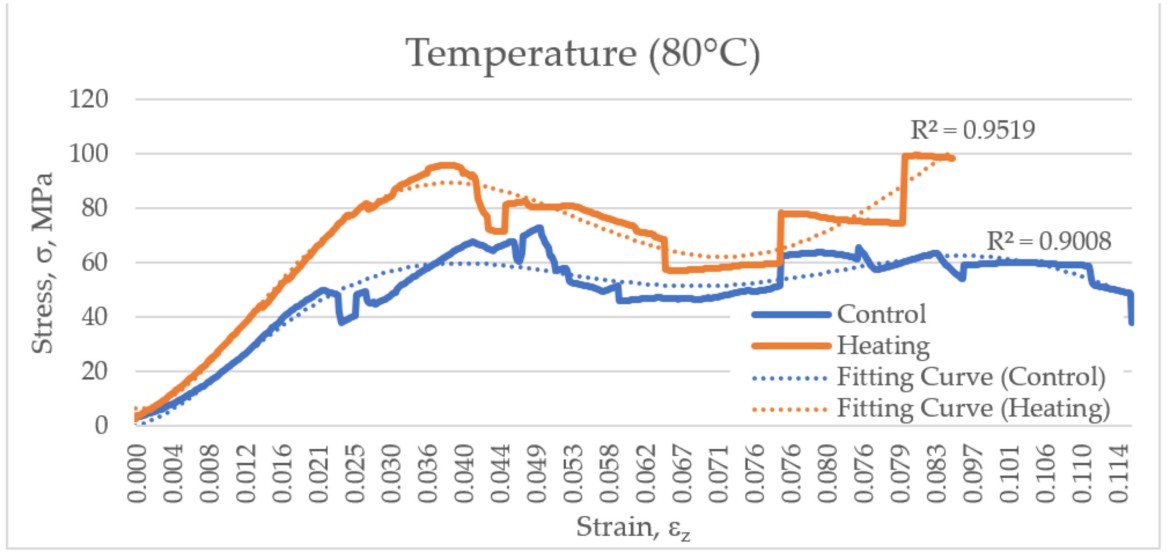

**Figure 6.** Bone samples' deformation curves following their heating to 80 °C.

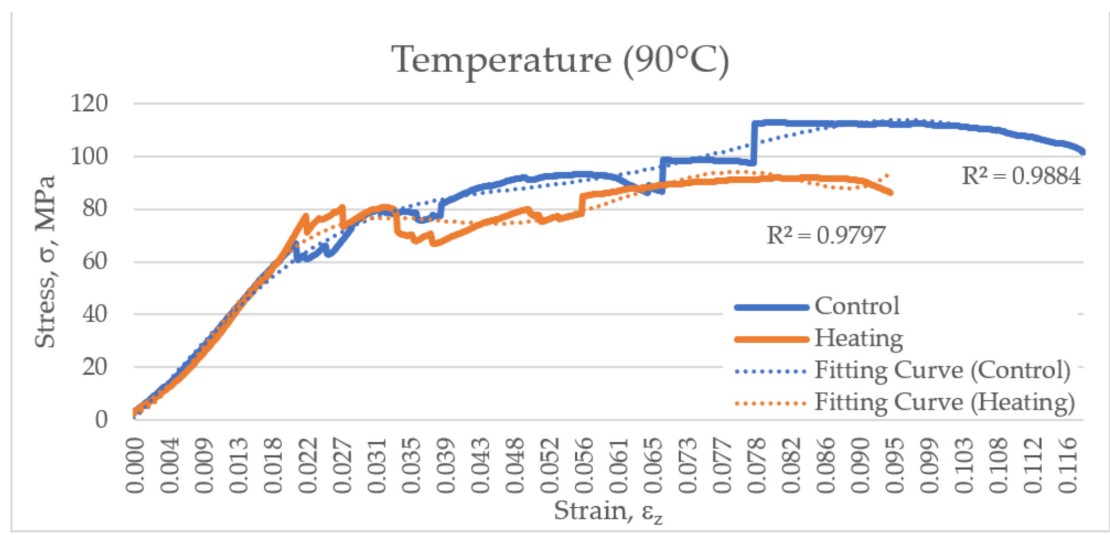

**Figure 7.** Bone samples' deformation curves following their heating to 90 °C.

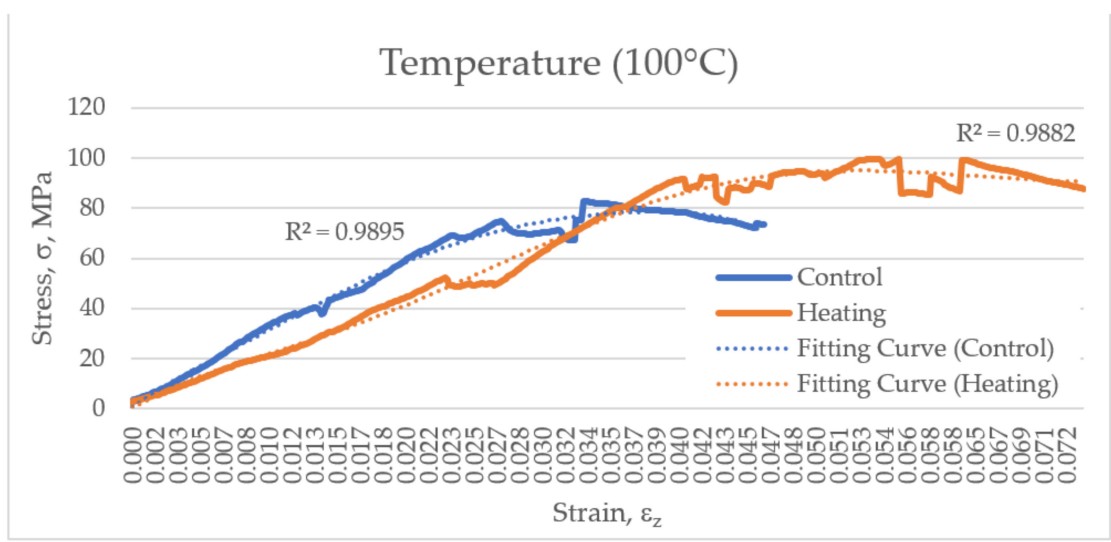

**Figure 8.** Bone samples' deformation curves following their heating to 100 °C.

There are variations in the stress-to-failure and strain-to-failure curves in the control (non-heated) group, which is typical for bone tissue due to its anisotropism. As shown in the diagrams, at 60 °C, the experimental and control curves are rather close (Figure 4). At 70 °C, rearrangement of the bone tissue under stress was noted (Figure 5). Interestingly, at 80 °C, bone strength against single-axis compression was higher than that in the control group ($p < 0.05$ as per Wilcoxon T-criterion); similar values of unit strain were noted at higher values of applied stress (Figure 6). The temperature of 80 °C may be an extreme point at which there are critical changes to the properties of strength and elasticity of bone tissue, as at 90–100 °C (Figures 7 and 8), the stress curve is located below the control curve, and at 100 °C, it moves into the plastic deformation area at a stress of 80 MPa (Figure 8). Some researchers argue that the temperature of 80 °C is critical to abrupt changes in properties of collagen [20].

Bones consist of an organic component, represented mainly by Type I collagen, and a mineral component, consisting of calcium phosphates, crystalline calcium hydroxyapatite, and tricalcium phosphate. Changes in the ratio between the mineral and bio-organic components of a bone are accompanied by changes in strength-related properties; a bone deprived

of organics becomes very brittle, while a demineralized bone takes on the properties of rubber [27–29].

Interpreting changes in the mechanical properties of biological objects, including bones, is a challenging task due to their composite or hybrid nature defined by the presence of various organic and inorganic components. Nevertheless, in order to assess the biomechanical processes taking place in bone samples during their single-axis compression following heating, individual strength-related characteristics were analyzed.

Having been subjected to hyperthermic treatment in the range of 60–100 °C, bone tissue samples endured a higher maximum compressive stress (Figure 9). However, for each separated heating point, no significant differences were discovered between the control and experimental groups (as per the Mann–Whitney test).

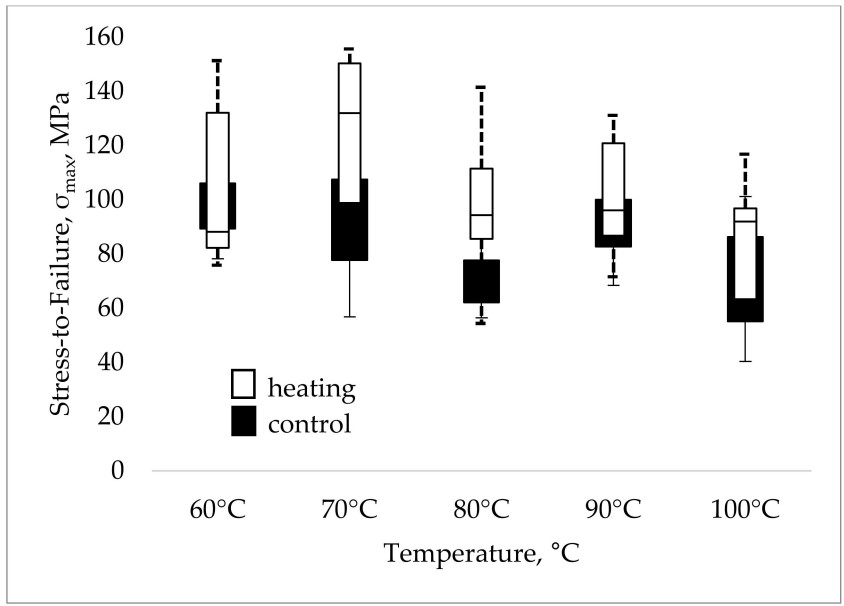

**Figure 9.** Diagram of changing stress-to-failure for various bone heating modes.

Hyperthermic treatment aims to raise the temperature of a target tissue to the range of 40–44 °C for at least 30–60 min to direct tumor damage. However, the full clinical potential of hyperthermic treatment occurs in synergy with chemotherapy and radiotherapy [2]. In this regard, the use of higher temperatures allows us to hope for a decrease in the doses of chemotherapeutic and radiotherapeutic procedures used to suppress tumor growth. At the same time, high temperatures carry an increased risk of impaired regeneration of healthy bone tissue. The obtained results are consistent with the data in [30], where the authors discovered that a 30 min microwave-induced hyperthermy of dog bone fragments in a range of 70–100 °C had no influence on the parameters of the ex vivo test for stress-to-failure and hardness of bone tissue. In a different report, the mechanical strength of a bone was reduced by 10% following its 30 min heating to a temperature of 70 °C; however, its maximum stress (stress-to-failure, in our case) did not change [31].

The compressive strength of a bone is largely defined by its density [32], which depends on the properties of thermally stable calcium salts. Elastic properties of bone tissue, e.g., under twisting or bending, are largely determined by bone collagen [33], which is a temperature-dependent protein, and as such is susceptible to temperatures above 56 °C. Thus, the next parameter for inter-group comparison of the strength properties of bone tissues was the strain-to-failure indicator. In other words, the moment when deformation becomes failure as a result of the stress was studied (Figure 10). This value covers elastic (Figure 11) and plastic (Figure 12) deformation and largely defines the value of the Young's modulus of elasticity (Figure 13).

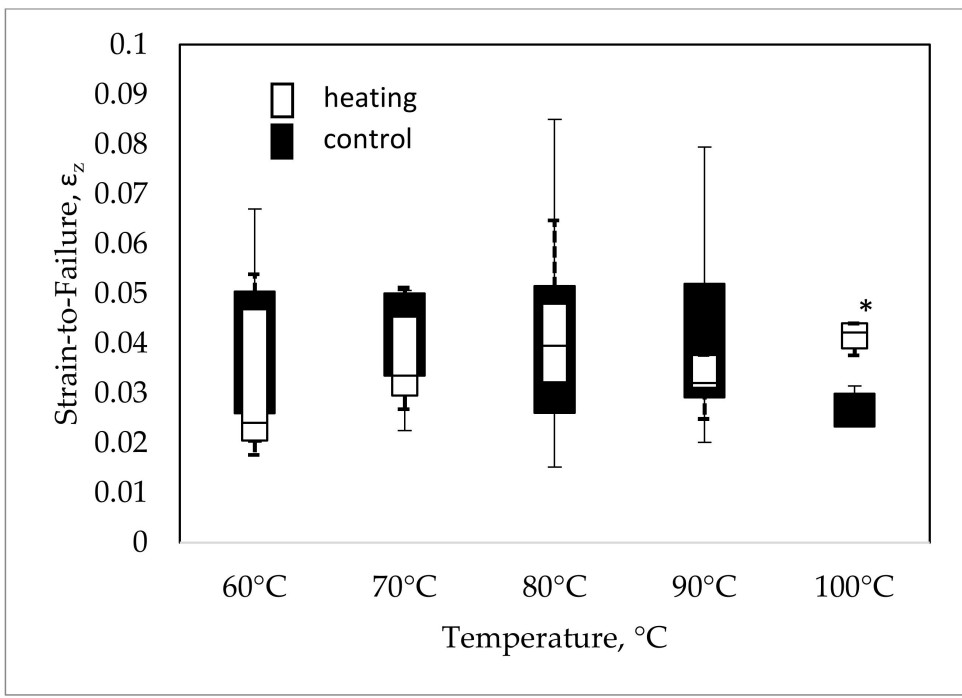

**Figure 10.** Diagram of bone strain for various bone heating temperatures. Asterisk denotes that the results are statistically significant as per the Mann–Whitney test.

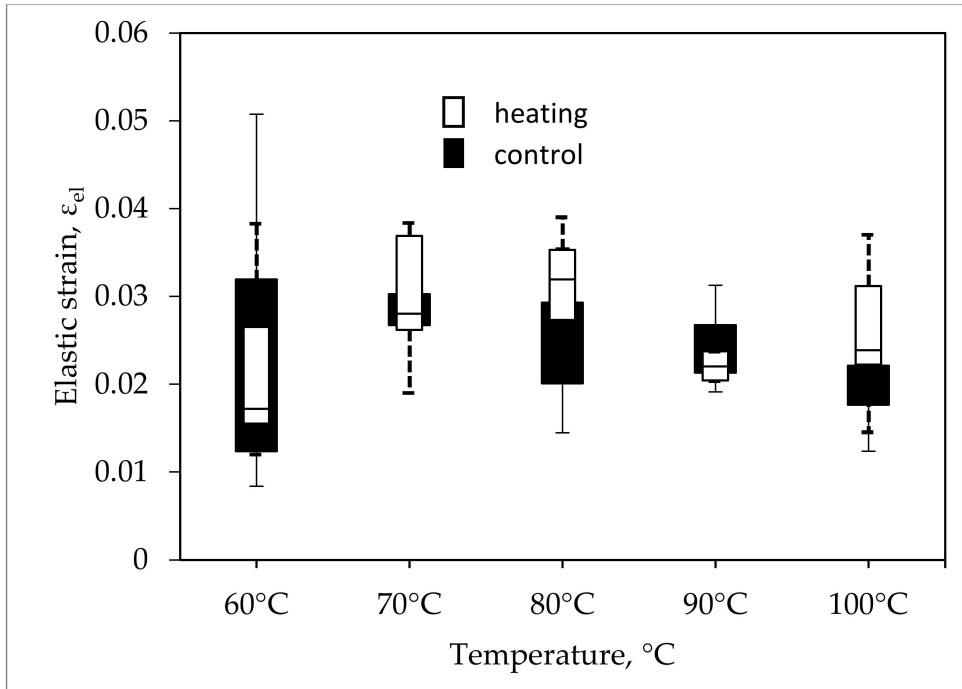

**Figure 11.** Diagram of elastic bone strain for various bone heating temperatures.

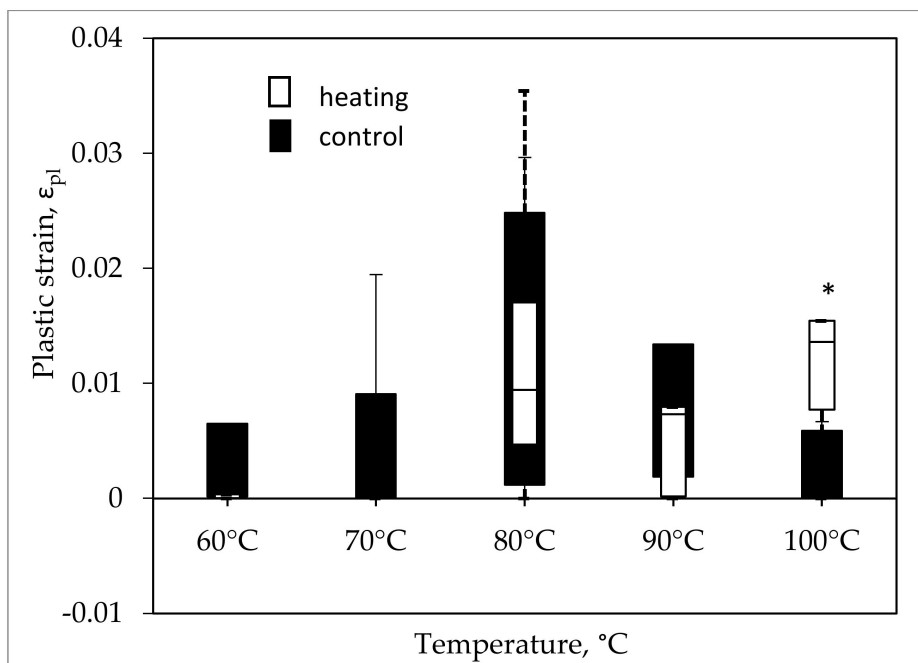

**Figure 12.** Diagram of plastic bone strain for various bone heating temperatures. Asterisk denotes the results are statistically significant as per the Mann–Whitney test.

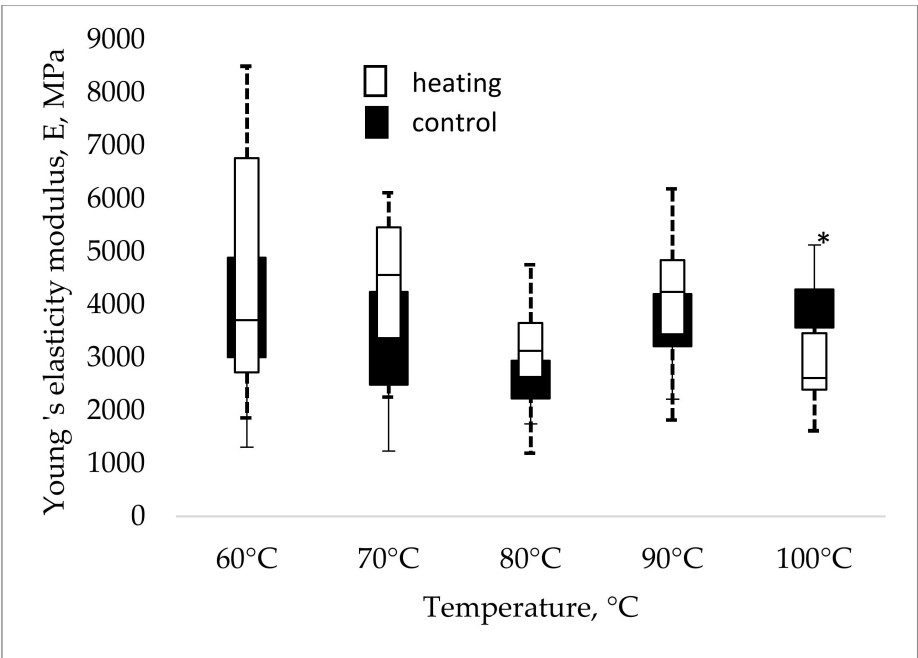

**Figure 13.** Diagram of modulus of elasticity of bone for various bone heating temperatures. Asterisk denotes that the results are statistically significant as per the Mann–Whitney test.

From Figures 10–13, we see that statistically significant changes in indicators with respect to controls were evident only at 100 °C. Under this temperature, unit strain increased by 68% ($p = 0.003$) of the control value (Figure 10), which was due to a four-fold increase ($p = 0.02$) in the plastic strain values (Figure 12). Plastic strain tended to zero-up to temperatures of 60–70 °C. When heating to 100 °C, the indicator of bone elastic strain increased only by 13% with a significance of $p = 0.063$.

It has been described in several research papers that ex vivo bone fragment heating temperatures in a range close to 60 °C have no practical effect on the properties of collagen

fibers [32,34]. The undulating nature of changes in indicator values leads us to suspect that, under gradual heat, collagen may change its state, transitioning to a sol-gel conformation as a result of the release or absorption of water molecules, thus suggesting periods of fluidity or brittleness of protein fibers. Following this change in state is a growing coagulative (i.e., dry) necrosis (denaturation) of bone cells and the destruction of collagen and other bone tissue proteins [35], leading to the shortening and the increased fragility of collagen fibrils, and resulting in a corresponding reduction in the elastic properties of the bone. According to our data, this process becomes critical to bone biomechanical properties at 100 °C, when the Young's modulus of pre-heated bone samples falls to 68% of the control level ($p = 0.034$, Figure 13).

## 4. Conclusions

Local hyperthermia over 50 °C may exert a therapeutic influence on pathological tissues that is favorable for the treatment of not only bone tumors, but also infections and inflammations [3]. For example, hyperthermic treatment at almost 55 °C had a high antibacterial efficiency (99.995%) against bone infections [36].

Although different methods of heating to different temperatures have varying positive and negative features, the issue of the effect of thermal exposure on bone strength is common to all heating methods and their variations. The study of the effect of heating on bone strength was the key to the present work.

An ex vivo study of thermoablation treatment was conducted on long tubular bones of healthy pigs involving periosteal placements of heating elements. In the case of a one-hour treatment in a temperature range of 60–90 °C, no significant changes in strength characteristics were detected.

Verifiable changes in the mechanical characteristics of bone tissue under compression, such as reduced modulus of elasticity and increased elastic strain (strain-to-failure), were detected when heating to 100 °C, mainly due to increases in plastic strain. This finding may serve as evidence of a critical ex vivo change in the biomechanical behavior of bone tissues at such temperatures. Prolonged (48 weeks) unsatisfactory biomechanical behavior of bones treated ex vivo at 100 °C for 5 min was previously demonstrated in [32]. This effect may be related to the thermal shrinkage of bone collagen, resulting in worsening biomechanical indexes of the treated bone [32] and its regenerative properties [34].

The conducted research has its limitations. The small sample size of animals, the limited number of bone samples used in biomechanical tests, and the presence of only one stress application scheme used in the ex vivo testing do not allow for a sound conclusion to be made. Nevertheless, the results imply the viability of further testing using the suggested hyperthermy technique (with bone tissue treatment to a temperature of 60–90 °C) for thermal ablation of tumor-involved bones ex vivo, in situ, and later, in vivo.

**Author Contributions:** Conceptualization, D.P. and V.P.; methodology, D.P. and A.K. (Alexander Kashin); validation, D.P., V.P. and I.A.; formal analysis, D.P. and A.K. (Alexey Kulkov); investigation, V.P. and I.S.; resources, V.P. and I.A.; data curation, V.P.; writing—original draft preparation, D.P., V.P. and I.K.; writing—review and editing, D.P., V.P., and I.K.; visualization, V.P.; supervision, E.K., D.P.; project administration, E.K.; funding acquisition, E.K. All authors have read and agreed to the published version of the manuscript.

**Funding:** This research was funded by the Ministry of Science and Higher Education of the Russian Federation within the framework of scientific projects carried out by teams of research laboratories of educational institutions of higher education subordinate to the Ministry of Science and Higher Education of the Russian Federation, project number FEWM-2020-0042. This research was supported in part by Siberian State Medical University development program Priority 2030.

**Institutional Review Board Statement:** Ethical review and approval were waived for this study due to using the pig femurs from a meat packing plant.

**Acknowledgments:** The authors would like to thank Irkutsk Supercomputer Center of SB RAS for providing the access to HPC-cluster Akademik V.M. Matrosov [37].

**Conflicts of Interest:** The authors declare no conflict of interest. The funders had no role in the design of the study; in the collection, analyses, or interpretation of data; in the writing of the manuscript; or in the decision to publish the results.

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
