# Peer review of "Compressive Strength Characteristics of Long Tubular Bones after Hyperthermal Ablation"

_symmetry, doi:10.3390/sym14020303_

Round 1

Reviewer 1 Report

The thermoablation is used as treatment of tumorous bones, but its influence on the mechanical properties is less analysed.  The authors proposed an ex vivo study of the dry thermal treatment influence in a range of 60-100℃ on structural properties of pig femur using an original approach. They were found useful correlations which can open a new approach in the thermoablation efficiency against tumor involvement of bones.

Even the authors underline their research limitations, in my opinion they opened a new and very useful direction in tumoral bones studies, by means of the presented hyperthermy technique (with bone tissue treatment to a temperature of 60-90℃) in application to thermal ablation to tumor-involved bones ex vivo, in situ, and later in vivo.

Some small suggestions:

The symbolisations by dotted lines of the polynomial approximations for heated and control specimens have to be shown in each figure (i.e. in Figures 3…7);

Also, the Russian words have to be removed from these figures in order to became more clear these useful diagrams.

The presented contribution is valuable and very useful.

I also suggest to the authors to take into the consideration as a further goal/direction to study the bone specimens’ cyclic loading (what is happened in fact with the real bone tissue).

Author Response

Thank you very much for your interest in our work.

1. The symbolisations by dotted lines of the polynomial approximations for heated and control specimens have to be shown in each figure (i.e. in Figures 3…7);

Author comments: All these figures contain the dotted lines of the polynomial approximations. May be there were problems with the orange lines.

2. Also, the Russian words have to be removed from these figures in order to became more clear these useful diagrams.

Author comments: You are right. It was corrected.

3. The presented contribution is valuable and very useful.

I also suggest to the authors to take into the consideration as a further goal/direction to study the bone specimens’ cyclic loading (what is happened in fact with the real bone tissue).

Author comments: Thank you, it may be the goal of our future work.

Reviewer 2 Report

Reviewers' comments:

The manuscript deals with the Compressive strength characteristics of long tubular bones after hyperthermal treatment. The rationale behind the work is clear, the state of the art is well reported and representative, characterization is comprehensive and dealt with care and statistical significance; experiments are well described. Results are clear and interpreted properly. The text flows nicely and linearly, all claims are supported by experiments and literature both.

Query:

Although the core of the work is solid, some support parts should be improved and some points of the text should be clarified: -

  1. The world “Hyperthermia” and “Thermal ablation” should be explain properly in manuscript. Is there any difference between these two words?
  2. In table 1. after heating 80 oC strain is reducing but, raising strain at 100 oC not explained properly. In similar way at 90 oC heating reduction in elastic and plastic deformation had highlighted but raising the value of elastic and plastic deformation after 100 oC need explanation.

3.     Is there any effect on the reported properties by varying the heating time?

4. Any microstructural or mathematical modelling and simulation studies may be added if available.

Author Response

Thank you very much for your interest in our work.

Although the core of the work is solid, some support parts should be improved and some points of the text should be clarified: -

  1. The world “Hyperthermia” and “Thermal ablation” should be explain properly in manuscript. Is there any difference between these two words?

Author comments: Thermal ablation and hyperthermia both are parts of hyperthermal treatment.

     2. In table 1. after heating 80 oC strain is reducing but, raising strain at 100 oC not explained properly. In similar way at 90 oC heating reduction in elastic and plastic deformation had highlighted but raising the value of elastic and plastic deformation after 100 oC need explanation.

Author comments: Sorry, table 1 was not used to compare statistical differences in the findings obtained at different heating temperatures. This table showed the deviations of indicators from Gaussian normal distribution law, only. In fact, Fig.9 demonstrated statistical raising the values of strain after 100°C heating caused by plastic deformation (Fig.11), mainly. The interpretation of this phenomenon is complex and practically not described in the literature. However, we tried to give an explanation on pages 7-8 (see references 31-34).

  1. Is there any effect on the reported properties by varying the heating time?

Author comments: There were no any experiment for understanding the influence of heating time on strength characteristics.

  1. Any microstructural or mathematical modelling and simulation studies may be added if available.

Author comments: There were no such modelling and simulation.

Reviewer 3 Report

General comment:

This work aims to investigate the influence of thermal treatment of bone tumors on healthy bone compressive strength. 
The manuscript is interesting. However, every section calls for a review and significant improvements. In particular, the methodological and results sections has to be greatly enhanced. Major flaws about the methods are evident. 

Specific comments throughout the paper:

A terrible flaw is immediately evident. The term hyperthermia is used in the wrong way. Hyperthermia is a thermal therapy whose aim is to rise the temperature of a target tissue in the range 41-45°C. Please see

10.1080/02656730118511

10.1016/S1040-8428(01)00179-2

10.1080/02656736.2020.1779357

10.1109/TBME.2021.3134208

As the temperature increase, protein coagulation occurs, water is lost and ABLATION occurs. Given that the temperature range the authors declares is 60-100°C, the title and maintext must be revised. 

1. Introduction 

Line 62-74: The authors are missing to report that hyperthermia (with lower temepratures) was selected as a preferred approach, because radiofrequency or microwave ablation performed withouth significance reduction in the recurrence rate, while providing relevant drawbacks in the post-operative stage. Please see: 10.1109/TBME.2021.3134208. This is why, in the literature, multi-functional, responsive materials (e.g., magnetic or light-responsive) are increasingly studied as innovative tool to perform thermal treatment directly at the tumor site, with the possibility of favoring the bone healing after the surgery. I suggest the athors to read and compare with

10.7150/thno.49784

10.1109/JMMCT.2019.2959585

The quality of the Introduction must be improved. 

2. Research Methods

I suggest the authors to think of a new figure capable of explaining the problems, methods and ais of their work, thus guiding the readers to a direct and immediate understanding of the value of their experiments and results.

Line 96: Missing reference(s) and technical details. The equipment should be described by providing information on the type of energy used for the heating, the type of applicator, the focal spot (zone of influence, in cm) and other. I suggest to follow the technical guidelines from:

10.1080/02656736.2016.1277791

10.1007/s00066-017-1106-0

10.1080/02656736.2018.1564155

Given that the Phoenix -2 local hyperthermy equipment is a direct current injector, the metallic Pt 100 thermometric probes can interact with the current field and modify the treatment? Discuss about the validity of your thermometric setup. 
Furthermore, tell how many thermometer did you use and where did you place them. 

Line 102: Why rectangular block? According to Fig. 2, the samples seems to be cylinidrical. Maybe I a missing something or this part is confusing. I suggest to revise it. 

Line 113: Not "hyperthermy" but "hypethermia treatment". Please revise.

Line 134: Some russian text was not translated or removed. Please fix. 

Lines 130-143: This part is a mixture of results and methods. I suggest to consider a re-organization. Instead, a coherent discussion is in order. 

3. Results and Discussion

The quality of Fig. 3-7 can be improved: i) the legends are overlapped (and some russian text remain) and the dot orange is not reported, ii) the strain is in % or not? 

Furthermore, how the fitting was performed? The fitting seems to present a relatively large percentage error. The authors are not presenting any figure of merit to evaluate the quality of the fitting. Please provide a R^2 or something which can help in assessing your results. 

The figures present a largely different x-axis, having a very different upper bound, respectively, 0.099, 0.202, 0.114, 0.117, 0.702. It is therefore difficult to compare when the maximum stress is occuring for the different cases. In other words, it is difficult to assess if some heat-treated bones present a more linear or non-linear stress-strain behavior. I suggest to use a unique x-axis.

Line 171-181: Did the author consider to fit the stres-strain relationship to some equivalent circuit model and then derive the parameters to quantitatively rule the depdenence of visco-elastic properties from the thermal treatment? If reference about the lumped models of bone mechanical response are needed, please ask. 

Lines 188-193: Please read the Introduction from 10.1109/TBME.2021.3134208 to find additional references related to bone damage and microwave thermal treatment. It is better to expand this rather crucial part. This can be a relevant finding for your study, please consider to compare with all available references to increase the quality and value of your work. 

The way the eleastic strain, the plastic strain and the Young's elasticity modulus were calculated from the stress-strain (Figs. 3-7) must be reported. Please revise the methodological section. 

Line 203-208: The authors, in my opinion, are neglecting to comment on some microscopical aspects related to the temperature-induced changes at the microscopic scale. I suggest them to read the following article, where the biological effects of ablation temperatures is clearly described:

10.1615/CritRevBiomedEng.2015012486

What holds for electromagnetic properties can be translated to the mechanical properties.
The fact that (Fig. 9, 11, 12) statistically significant changes are occuring only for  bone heat treated at 100°C must be explained clearly, because all endpoints behave in the same way and the authors must prove that this is not a case or an error.  

Line 212: missing reference. 

4. Conclusions

Conclusions can be expanded by providing a discussion with an interdisciplinary persepctive. For instance, I suggest the authors to consider to comment about how their methods and results can be relevant for biomaterial-based approaches for treating bone tumors: 

10.7150/thno.49784

10.1109/JMMCT.2019.2959585

10.1109/TBME.2021.3134208

It is also possible that the design of a functional biomaterial can be driven by your results. 
Please enhance the quality of the Conclusion section. 

Author Response

Thank you very much for your interest in our work.

We tried to take into account your comments in the new version of the work and give answers to the questions asked. The file with the answers is attached below, the number of lines within the answers is according to the new work file.

Thank you very much for your valuable advice.

Round 2

Reviewer 2 Report

The manuscript deals with the Compressive strength characteristics of long tubular bones after hyperthermal treatment. The reviewer is satisfied with the clarification submitted by the authors. Hence, the manuscript can be accepted for publication.  

Author Response

The manuscript deals with the Compressive strength characteristics of long tubular bones after hyperthermal treatment. The reviewer is satisfied with the clarification submitted by the authors. Hence, the manuscript can be accepted for publication. 

Reviewer 3 Report

I thank the authors for the time they spent in revising their work and in replying to my questions and doubts.

The authors put a lot of efforts in modifying the manuscript according to my comments. However, some issues still has to be fixed. 

Line 3: in the title the first letters are capital. Please fix.

The Introduction section was not improved and revised. This is a lack and lowers the quality of the work. The authors replies (e.g., the comparison between hyperthermia and thermoablation against bone tumors) have not translated into modifications in the text. I sugget to revise the Introduction by relying on your reply to my doubts.

As regards the requested techinical details about the heating apparatus (around line 70), the authors are recalling some references which are very difficult to find. Some of the explanations about the heating element, taken from [10], can be added. In this way, the authors can explicitely explain the type of heat administration they have performed and avoid misunderstandings. 

Thank you for the explanation of the stress calculations. 

In Fig. 3 some graphical elements requires to be revised and modified. Indeed, the newly added labels and symbols has the "¶" which suggest that a screenshot was taken in some text editor. Please revise Fig. 3 to improve its clarity and quality. Furthermore, on the x-axis the label is not similar to the other stress-strain plots, and the decimals are expressed with the coma (",") instead of using the dot ("."), as done in the other figures Please fix. 

There is a gap, from a conceptual point of view, between the newly added lines 275-278 and the old part of the Conclusions. A review is required at this point to get a fluent and coherent text. 

Author Response

Thank you very much for your comments, we've taken it into account.

Line 3: in the title the first letters are capital. Please fix.

The design of English-language titles is not very clear to us, however, the format of titles of the MDPI publishing house implies the use of first capital letters in titles, except for prepositional words (for example, https://www.mdpi.com/2073-8994/14/1).

The Introduction section was not improved and revised. This is a lack and lowers the quality of the work. The authors replies (e.g., the comparison between hyperthermia and thermoablation against bone tumors) have not translated into modifications in the text. I sugget to revise the Introduction by relying on your reply to my doubts.

Thank you, corrected.

As regards the requested techinical details about the heating apparatus (around line 70), the authors are recalling some references which are very difficult to find. Some of the explanations about the heating element, taken from [10], can be added. In this way, the authors can explicitely explain the type of heat administration they have performed and avoid misunderstandings.

Thank you very much, technical information about the Phoenix-2 complex has been added

Thank you for the explanation of the stress calculations.

In Fig. 3 some graphical elements requires to be revised and modified. Indeed, the newly added labels and symbols has the "¶" which suggest that a screenshot was taken in some text editor. Please revise Fig. 3 to improve its clarity and quality. Furthermore, on the x-axis the label is not similar to the other stress-strain plots, and the decimals are expressed with the coma (",") instead of using the dot ("."), as done in the other figures Please fix.

Thank you, corrected.

There is a gap, from a conceptual point of view, between the newly added lines 275-278 and the old part of the Conclusions. A review is required at this point to get a fluent and coherent text.

Thank you very much, a smoother semantic transition has been added.

Round 3

Reviewer 3 Report

Thank you for revising your work and for modifying it to include your replies in the Introduction and in the Methodological sections, as well as for the addition in the Conclusion section. The quality of the paper has increased. 

No further comments from me.